# Application of Ensemble Algorithm Based on the Feature-Oriented Mean in Tropical Cyclone-Related Precipitation Forecasting

**Jing Zhang and Hong Li ***

Shanghai Typhoon Institute, China Meteorological Administration, Shanghai 200030, China; zhangj@typhoon.org.cn
* Correspondence: lih@typhoon.org.cn

**Abstract:** Tropical cyclones (TCs) are characterized by robust vortical motion and intense thermodynamic processes, often causing damage in coastal cities as they result in landfall. Accurately estimating the ensemble mean of TC precipitation is critical for forecasting and remains a foremost global challenge. In this study, we develop an ensemble algorithm based on the feature-oriented mean (FM) suitable for spatially discrete variables in precipitation ensembles. This method can adjust the locations of ensemble precipitation fields to reduce the location-related deviations among ensemble members, ultimately enhancing the ensemble mean forecast skill for TC precipitation. To evaluate the feasibility of the FM in TC precipitation ensemble forecasting, 18 landing TC cases in China from 2019 to 2021 were selected for validation. For precipitation forecasts of the landing TCs with a varying leading time, we conducted a comprehensive quantitative evaluation and comparison of the precipitation forecast skills of the FM and arithmetic mean (AM) algorithms. The results indicate that the field adjustment algorithm in the FM can effectively align with the TC precipitation structure and the location of the ensemble mean, reducing the spatial divergence among precipitation fields. The FM method demonstrates superior performance in the equitable threat score, probability of detection, and false alarm ratio compared with the AM, exhibiting an overall improvement of around 10%. Furthermore, the FM ensemble mean shows a higher pattern of the correlation coefficient with observations and has a smaller root mean square error than the AM ensemble mean, signifying that the FM method can better preserve the characteristics of the precipitation structure. Additionally, an object-based diagnostic evaluation method was used to verify forecast results, and the results suggest that the attribute distribution of FM forecast objects more closely resembles that of observed precipitation objects (including the area, longitudinal and latitudinal centroid locations, axis angle, and aspect ratio).

**Keywords:** ensemble mean; field alignment; tropical cyclone; precipitation forecasts; multisource fusion precipitation

## 1. Introduction

Tropical cyclones (TCs) are cyclonic vortices occurring over warm tropical ocean regions characterized by warm core structures. They exhibit intense vortical motion and thermodynamic processes, such as vigorous moist convection, frequently leading to destructive weather, including strong winds, heavy rainfall, and storm surges upon landfall in coastal cities. According to the Emergency Events Database (EM-DAT; https://data.humdata.org/dataset/philippines-typhoon-impact-data-2014-2020), TCs constituted the largest global natural disaster in 2019, affecting over 30 million people and causing direct economic losses exceeding USD 50 billion.

The western North Pacific (WNP) is one of the most TC-prone areas in the world, with approximately 36% of TCs occurring there every year on average. China, located on the western rim of the WNP, with a coastline extending over 18,000 km, ranks among

the most severely and frequently affected areas by TC, and storm surge disasters. The TCs originating in the WNP (including the South China Sea) amount to approximately 27 per year, with about 16 affecting China per year and about 7 making landfall per year [1], resulting in annual direct economic losses of up to USD 9 billion for China [2]. Moreover, with the rapid population and economic growth in the coastal areas of eastern China, the losses caused by TC disasters continue to increase, and the risk of TC disasters grows [3]. Consequently, research on TC disasters has gained widespread attention [4,5]. The landing TCs in China not only bring extreme weather phenomena, such as heavy rainfall, strong winds and storm surges locally, but also seriously affect inland areas, posing a serious threat to the safety of people's lives and property by causing urban waterlogging and flooding. Therefore, accurately forecasting the heavy rainfall caused by landing TCs is crucial for disaster prevention and mitigation. However, due to the complex multiscale dynamic and thermodynamic processes, as well as the influence of the landform and boundary condition in the typhoon landfall area, forecasting the intensity and location of landing TC precipitation still faces considerable uncertainty, which is a formidable international challenge.

Furthermore, there are still several objective problems, such as the intricate interactions of dynamic and thermodynamic processes of the TCs at multiple spatio-temporal scales, the shortcomings of the framework and physical processes of current forecasting models, the incompleteness of observation systems specific to the TCs, and issues with assimilation algorithms and systems for high-resolution models. Therefore, large errors remain in single deterministic forecasts for the location and intensity of TC precipitation, even in the most advanced assimilation/forecast systems [6–8]. Moreover, deterministic forecasts are unable to quantitatively estimate the uncertainty of forecast results. Given these challenges, an increasing number of international research and operational institutions have adopted ensemble forecasting to quantitatively predict TC precipitation, i.e., using ensemble averaging, ensemble spread, and the probability to perform quantitative estimates.

Among these metrics, ensemble averaging has been widely applied due to its simplicity in calculation and effectiveness in extracting predictable information to reduce forecast errors. The track and intensity of a TC include scalar quantities, and their ensemble averages can be obtained through a straightforward calculation of the average values of TC track and intensity among ensemble forecast members (a consensus forecast). Numerous research findings have demonstrated that ensemble mean forecasts remarkably improve the forecast skills for the TC track and intensity compared with single deterministic forecasts [9–13]. Nevertheless, unlike TC track and intensity, TC precipitation is a spatial variable, making its forecasting inherently unique. On the one hand, the location deviations of TC centers among ensemble members increase with the forecast leading time, which inevitably leads to substantial deviations in the TC rainfall area among ensemble members. On the other hand, TC precipitation mainly originates from the inner rainbands caused by eyewall vortices and the outer rainbands associated with outflow layers. These precipitation processes show a large uncertainty due to the complexity of internal physical processes in vortex-relative coordinates. This further exacerbates the deviations in TC precipitation intensity and locations among ensemble members. The distinctive nature of TC precipitation poses challenges for the application of the arithmetic mean (AM) algorithm, which is a simple ensemble averaging method based on point-wise averaging, in TC precipitation ensemble forecasts. Specifically, the AM algorithm tends to overlook differences in precipitation locations, which can result in the false smoothing of precipitation fields and the deformation of precipitation structures. This smoothing effect often causes AM forecasts to underestimate TC precipitation intensity but overestimate the TC precipitation range. Furthermore, some scholars shifted TC precipitation structures based on the TC center position and then calculated the average values [14–18]. While this approach partly mitigates the problems associated with the first aspect mentioned above, it still neglects the impact of the second aspect on TC precipitation forecasts. The limitations

of the AM method in precipitation ensemble average forecasts largely restrict the broader, deeper, and more effective application of ensemble forecasting methods.

Although this issue in TC precipitation ensemble forecasting has long been recognized, a satisfactory solution remains elusive. [18] proposed a novel perspective to enhance TC ensemble forecasting skills by reducing the location deviations of precipitation structures among ensemble members. This ensemble algorithm, based on the feature-oriented mean (FM), adjusts the similar structural features of different ensemble forecasting fields to their mean position before calculating the amplitude average. The results indicate that the FM method has demonstrated superior performance in continuous variables. However, since the precipitation pattern is a spatially discrete variable, the applicability of the FM method in TC precipitation ensemble forecasting warrants further investigation. Therefore, in this study, we aim to develop the FM method and investigate its feasibility in TC precipitation ensemble average forecasting. In addition, the effectiveness of the FM in improving TC precipitation forecast skills is comprehensively evaluated.

In this study, we specifically investigate the following questions:

(1) Considering the spatial non-continuity of TC precipitation, can an FM algorithm be developed for TC precipitation fields, and how can it be effectively employed to regionally adjust precipitation fields, reducing deviations in TC precipitation locations among ensemble members?

(2) To what extent can the FM method improve the forecast skill of TC ensemble mean precipitation compared to the traditional AM method? How does this improvement vary with the leading time of ensemble forecasts?

The remainder of this paper is arranged as follows. Section 2 provides the methods and experimental data used in this research. Section 3 presents the experimental design and a selection of experimental samples. Section 4 introduces the verification methods for the forecast results, including the traditional "point-to-point" verification method and the spatial verification method for object-based diagnostic evaluation (MODE). Section 5 shows the forecast performance of the FM method for TC precipitation forecasts in the Global Ensemble Forecast System (GEFS) of the National Centers for Environmental Prediction (NCEP). The main conclusions and discussion are summarized in Section 6.

## 2. Methods and Experimental Data

### 2.1. Feature-Oriented Ensemble Mean (FM) Algorithm

In recent years, we have proposed a novel FM algorithm to address the problems associated with traditional AM ensemble averaging [18]. Unlike the AM method, the FM algorithm, before calculating the ensemble average, adjusts the structural features of each ensemble forecast field to their mean positions, thereby reducing location deviations among the ensemble members. Specifically, the FM method consists of the following four steps (Figure 1).

Firstly, the field adjustment algorithm in the FM is used to compute the displacement vectors $\overrightarrow{D_{j,1}}$, $\overrightarrow{D_{j,2}}$,..., $\overrightarrow{D_{j,N}}$ between any ensemble member $j$ and every other ensemble member. The field adjustment algorithm, initially proposed, developed, and applied by [19] for fluid dynamics research, was enhanced by incorporating a spatial-scale restriction module. This module allows structural position adjustments at a specific scale and smooths the remaining scales, greatly improving the computational efficiency and applicability to variable atmospheric fields [18]. The algorithm is expressed by Equation (1).

$$J(X,q) = \frac{1}{2}\delta X^T C\left(X^f \cdot q\right)^{-1} \delta X + \frac{1}{2}\delta Y^T R^{-1} \delta Y + \bigwedge(q) \tag{1}$$

where $J(X,q)$ represents the cost function concerning the gridded scalar field $X$ after the adjustment and the displacement vector field $q$. The objective of the cost function is to measure the similarity of a solution $X$ in appearance to the prior estimate $X^f$ and in geometry to the second field $Y$. In our application, $X^f$ and $Y$ represent the original field,

and the target reference field, respectively. The structure of $X$ is similar to that of the original field $X^f$, but there are some location deviations. The core idea of the field adjustment algorithm is to minimize the cost function $J(X, q)$ to estimate the adjusted field $X$ and the displacement vector field $q$. This minimization ensures that, under certain constraints of the displacement vector $\bigwedge(q)$, $X$ has the least total fitted variance with the position-adjusted $X^f$ field and the target reference field $Y$. Here, $(X^f \cdot q)$ represents the field after $X^f$ moves along the displacement field at $q$. $\delta X = \left[ X - X^f \cdot q \right]$ and $\delta Y = [Y - X]$. $C(X^f \cdot q)$ and $R$ indicates the error covariance matrixes of $(X^f \cdot q)$ and the target field, respectively.

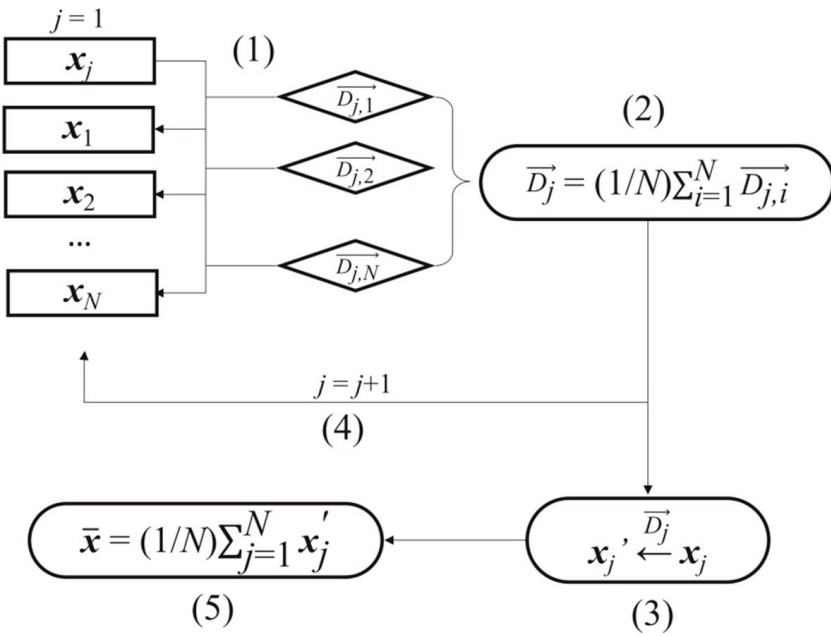

**Figure 1.** Flowchart of the feature-oriented mean (FM) method [18].

In the cost function, a constraint term $\bigwedge(q)$ regarding the displacement vector $q$ is introduced, which allows the field adjustment algorithm to be performed for features above a specific scale with smooth smaller-scale features. This dramatically improves computational efficiency [18]. The displacement vector $q$ obtained from Equation (1) represents the needed displacement vectors $\overrightarrow{D_{j,1}}$, $\overrightarrow{D_{j,2}}$, ..., $\overrightarrow{D_{j,N}}$.

Then, all ensemble members are moved to the mean position, as shown in Figure 2. Specifically, assuming three ensemble forecast members, we first use the field adjustment algorithm to estimate the displacement vectors between each ensemble member. The displacement vectors of Member 1 relative to Members 2 and 3 are denoted as $\overrightarrow{D_{1,2}}$ and $\overrightarrow{D_{1,3}}$, respectively. Therefore, the displacement vector of Member 1 relative to the mean position can be roughly estimated as the red arrow $\overrightarrow{D_1}$ in Figure 2, i.e., $\overrightarrow{D_1} = \left( \overrightarrow{D_{1,2}} + \overrightarrow{D_{1,3}} \right) / 3$. Similarly, the displacement vectors of Members 2 and 3, relative to the mean position, are denoted as $\overrightarrow{D_2}$ and $\overrightarrow{D_3}$, respectively. Subsequently, the fields of the three ensemble members are adjusted to the mean position based on their respective displacement vectors, $\overrightarrow{D_1}$, $\overrightarrow{D_2}$, and $\overrightarrow{D_3}$. By analogy, if there are $N$ ensemble forecast members, the $j$th ensemble forecast member $x_j$ is moved to the mean position according to $\overrightarrow{D_j} = (1/N)\sum_{i=1}^{N} D_{j,i}$, becoming the adjusted member $x_j'$.

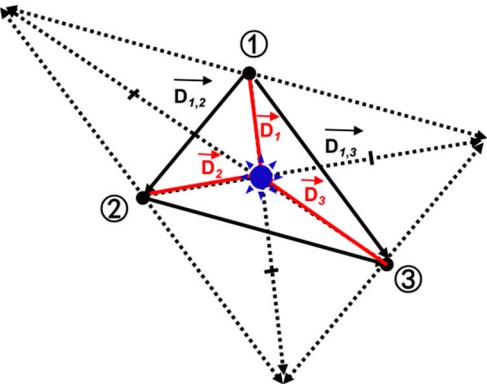

**Figure 2.** Schematic illustration of the movement of each ensemble member to the mean position through the FM method.

Thirdly, the above steps are repeated for each ensemble forecast member, i.e., $j = j + 1$. Thereby, all $N$ ensemble members are sequentially shifted to their mean positions.

Finally, the ensemble average for the shifted $N$ ensemble members ($x'_j$) is calculated, yielding the FM mean field ($\bar{x}$). It should be noted that in the FM method, adjustments are conducted separately for each grid point in the entire field. Thus, the distance and direction of movement are different for each grid point.

Note that, in contrast to the overall consistent movement for vortex repositioning, the FM employs a region-dependent adjustment for each ensemble member field. In other words, if a large (or small) location deviation exists in a specific region among the ensemble members, a corresponding large (or small) positional adjustment is applied. The field adjustment of the FM method is achieved by minimizing the cost function of the spatial field fitting through a variational algorithm.

*2.2. Data*

Data of the model ensemble forecast used in this study are from the National Centers for Environmental Prediction-Global Ensemble Forecast System (NCEP-GEFS) product, focusing on all landfalling TC cases in China from July to October in 2019–2021 (a total of 18 cases). The NCEP-GEFS product comprises 20 ensemble forecast members, with four forecasts per day at 00:00 UTC, 06:00 UTC, 12:00 UTC, and 18:00 UTC. The forecast variables include sea-level pressure, geopotential height, wind field, temperature field, precipitable water, and 6 h accumulated precipitation. The vertical levels include 250 hPa, 500 hPa, 850 hPa, and 1000 hPa, and the horizontal resolution is 0.5° × 0.5°.

To evaluate the forecasting performance of the FM method, the precipitation data for verification in this research used the precipitation fusion product from the China Meteorological Administration (CMA) Multisource Precipitation Analysis System. This precipitation product integrates ground-based observations, satellite-retrieved data, and radar-based quantitative precipitation estimates using key technologies such as bias correction and fusion analyses. The product covers the region of China (0–60°N, 70–140°E), with a horizontal resolution of 0.05° × 0.05° (regular latitude–longitude grid).

**3. Experimental Design**

This study aims to improve the forecast skill of land precipitation caused by TCs that result in landfall. Therefore, in several cases, the verification time for precipitation forecasts is fixed to assess the TC precipitation forecast skill at different leading times. Super Typhoon Lekima (1909) is taken as an example to introduce the experimental design of this study (Figure 3). Typhoon Lekima resulted in landfall in the coastal area of Chengnan Town, Wenling City, Zhejiang Province, China, at 01:00 Beijing Time (BJT) on 10 August 2019 (TC information in Table 1). Therefore, the verification time was chosen as the forecast time closest to this moment (00:00 UTC on 10 August 2019), and the accumulated precipitation of the landing TCs during the 6 h before this time (from 18:00 UTC on 9 August 2019 to

00:00 UTC on 10 August 2019) was used as the forecast object. Then, the precipitation forecast skills for different leading times were assessed. The forecast initiation times were set at 00:00 UTC on 7 August 2019, at 00:00 UTC on 8 August 2019, and at 00:00 UTC on 9 August 2019, corresponding to the leading times of 72 h, 48 h and 24 h, respectively.

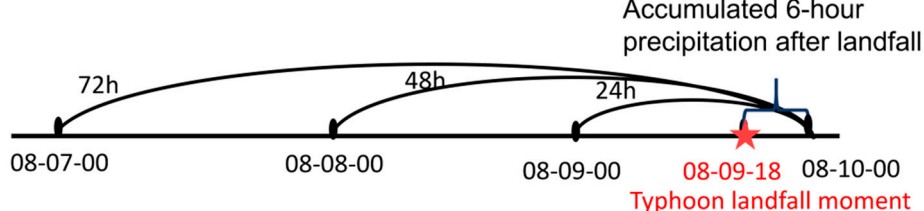

**Figure 3.** Schematic diagram of the experimental design, using super typhoon Lekima (1909) as an example.

**Table 1.** Information on the landing TCs in China in 2019, "BJT" indicates the Beijing Time, and "UTC" represents the Coordinated Universal Time.

| International Number | Name | Intensity | First Landfall in China | | | Model Forecast Time (UTC) |
|---|---|---|---|---|---|---|
| | | | Landfall Time (BJT) | Landfall Time (UTC) | Landfall Location | |
| 1904 | Mun | Tropical storm | 00:45 on 3 July | 18:00 on 2 July | Wanning City, Hainan Province | 00:00 on 2 July (24 h); 00:00 on 1 July (48 h); 00:00 on 30 June (72 h) |
| 1907 | Wipha | Tropical storm | 02:15 on 1 August | 18:00 on 31 July | Wenchang City, Hainan Province | 00:00 on 31 July (24 h); 00:00 on 30 July (48 h); 00:00 on 29 July (72 h) |
| 1909 | Lekima | Super typhoon | 01:45 on 10 August | 18:00 on 9 August | Wenling City, Zhejiang Province | 00:00 on 9 August (24 h); 00:00 on 8 August (48 h); 00:00 on 7 August (72 h) |
| 1911 | Bailu | Severe tropical storm | 13:00 on 24 August | 06:00 on 24 August | Pingtung County, Taiwan Province | 12:00 on 23 August (24 h); 12:00 on 22 August (48 h); 12:00 on 21 August (72 h) |
| 1914 | Kajiki | Tropical storm | 10:40 on 2 September | 00:00 on 2 September | Wanning City, Hainan Province | 06:00 on 1 September (24 h); 06:00 on 31 August (48 h); 06:00 on 30 August (72 h) |
| 1918 | Mitag | Typhoon | 20:20 on 1 October | 12:00 on 1 October | Zhoushan City, Zhejiang Province | 18:00 on 30 September (24 h); 18:00 on 29 September (48 h); 18:00 on 28 September (72 h) |

Figure 4 presents all 18 TCs that resulted in landfall in China during 2019–2021. These TCs primarily landed in South China and East China. The TCs landing in South China were typically generated in the waters east of the Philippines. Driven by southeasterly wind on the south side of the subtropical high and the north side of the monsoon trough, these TCs moved northwestward, landed in Taiwan through the Bashi Channel, and then crossed the Taiwan Strait to approach the eastern Guangdong Province or the coastal areas of Fujian Province. Eventually, they led to landfall on the coasts of Taiwan, Fujian, Guangdong, and the surrounding areas. If the TC originates from a relatively higher latitude, they cross the Ryukyu Islands and land on the coasts of Zhejiang Province, Shanghai, Jiangsu Province, and even Shandong Province and Liaoning Province. As the landfall locations of these TCs are usually in the more developed coastal cities of China, the economic and property losses caused by landfalling TC precipitation are huge. Improving the ensemble average forecasts of heavy rainfall caused by landing TCs is the central goal of this study.

## Best tracks of the TCs causing landfall in China during 2019–2021

**Figure 4.** The 6 h positions of tropical cyclones (TCs) in the western North Pacific (WNP, including the South China Sea, north of the equator and west of 180°E) during 2019–2021 (China Meteorological Administration Tropical Cyclone Best Track Dataset).

The information on all TCs that made landfall in China during 2019–2021 can be found in Tables 1–3, covering details such as TC number, TC name, TC intensity level, and the time and location of the TCs' first landfall in China. Additionally, the last column in Tables 1–3 show the forecast starting time and forecast leading time selected for each TC case in the experiments.

**Table 2.** Same as Table 1, but for the year 2020.

| International Number | Name | Intensity | First Landfall in China | | | Model Forecast Time (UTC) |
| --- | --- | --- | --- | --- | --- | --- |
| | | | Landfall Time (BJT) | Landfall Time (UTC) | Landfall Location | |
| 2002 | Nuri | Tropical storm | 08:50 on 14 June | 06:00 on 14 June | Hailing Island, Guangdong Province | 12:00 on 13 June (24 h); 12:00 on 12 June (48 h); 12:00 on 11 June (72 h) |
| 2003 | Sinlaku | Tropical storm | 07:15 on 1 August | 00:00 on 1 August | Wanning City, Hainan Province | 06:00 on 31 July (24 h); 06:00 on 30 July (48 h); 06:00 on 29 July (72 h) |
| 2004 | Hagupit | Strong typhoon | 03:30 on 4 August | 00:00 on 4 August | Leqing City, Zhejiang Province | 06:00 on 3 August (24 h); 06:00 on 2 August (48 h); 06:00 on 1 August (72 h) |
| 2006 | Mekkhala | Typhoon | 07:30 on 11 August | 00:00 on 11 August | Zhangpu County, Fujian Province | 06:00 on 10 August (24 h); 06:00 on 9 August (48 h); 06:00 on 8 August (72 h) |
| 2007 | Higos | Typhoon | 05:50 on 19 August | 00:00 on 19 August | Zhuhai City, Guangdong Province | 06:00 on 18 August (24 h); 06:00 on 17 August (48 h); 06:00 on 16 August (72 h) |
| 2016 | Nangka | Severe tropical storm | 19:35 on 13 October | 12:00 on 13 October | Qionghai City, Hainan Province | 18:00 on 12 October (24 h); 18:00 on 11 October (48 h); 18:00 on 10 October (72 h) |

**Table 3.** Same as Tables 1 and 2, but for the year 2021.

| International Number | Name | Intensity | First Landfall in China | | | Model Forecast Time (UTC) |
| --- | --- | --- | --- | --- | --- | --- |
| | | | Landfall Time (BJT) | Landfall Time (UTC) | Landfall Location | |
| 2104 | Koguma | Tropical storm | 09:45 on 12 June | 06:00 on 12 June | Lingshui City, Hainan Province | 12:00 on 11 June (24 h); 12:00 on 10 June (48 h); 12:00 on 9 June (72 h) |
| 2106 | In-fa | Strong typhoon | 12:30 on 25 July | 06:00 on 25 July | Zhoushan City, Zhejiang Province | 12:00 on 24 July (24 h); 12:00 on 23 July (48 h); 12:00 on 2 July (72 h) |
| 2107 | Cempaka | Typhoon | 21:50 on 20 July | 18:00 on 20 July | Yangjiang City, Guangdong Province | 00:00 on 20 July (24 h); 00:00 on 19 July (48 h); 00:00 on 18 July (72 h) |
| 2109 | Lupit | Tropical storm | 11:20 on 5 August | 06:00 on 5 August | Shantou City, Guangdong Province | 12:00 on 4 August (24 h); 12:00 on 3 August (48 h); 12:00 on 2 August (72 h) |
| 2117 | Lionrock | Tropical storm | 22:40 on 8 October | 18:00 on 8 October | Qionghai City, Hainan Province | 00:00 on 8 October (24 h); 00:00 on 7 October (48 h); 00:00 on 6 October (72 h) |
| 2118 | Kompasu | Typhoon | 15:20 on 13 October | 12:00 on 13 October | Qionghai City, Hainan Province | 18:00 on 12 October (24 h); 18:00 on 11 October (48 h); 18:00 on 10 October (72 h) |

## 4. Forecast Verification Methods

In this study, a comprehensive evaluation and comparison of TC precipitation forecast skills between the FM method and other ensemble averaging methods are conducted using different metrics. The first category comprises the "point-to-point" metrics, such as ensemble spread, root mean square error (RMSE) and pattern correlation coefficient (PCC), and binary classification metrics like the equitable threat score (ETS), probability of detection (POD), false alarm ratio (FAR) and index of agreement (IOA). This category is used to evaluate the performance of overall precipitation forecasts. The second category is object-based verification metrics, such as the object-based diagnostic evaluation (MODE) for spatial precipitation patterns.

### 4.1. Traditional Point-to-Point Verification

(1) The ensemble spread is calculated as follows:

$$\text{Spread} = \sqrt{\frac{1}{m}\sum_{i=1}^{m}\frac{1}{n-1}\sum_{j=1}^{n}\overline{\left(\overline{f}-f(j)\right)^2}} \tag{2}$$

where $\overline{f} = \frac{1}{n}\sum_{j=1}^{n}f(j)$ represents the ensemble mean value, $n$ is the number of ensemble members, and $m$ is the total number of samples. Ensemble spread is used to measure the uncertainty of the ensemble forecast system.

(2) The root mean square error is obtained as follows:

$$\text{RMSE} = \sqrt{\frac{1}{m}\sum_{i=1}^{m}(f_i - o_i)^2} \tag{3}$$

RMSE can be used to represent forecast accuracy. $m$ denotes the total number of samples, and $f_i$ and $o_i$ indicate the forecasted and observed values for the $i$th sample, respectively. The RMSE can gauge whether an ensemble forecast system exhibits a reasonable spread by comparing it with the ensemble spread.

(3) The pattern correlation coefficient is calculated as follows:

$$\text{PCC} = \frac{\frac{1}{N}\sum_{i=1}^{N}\left(f_i - \overline{f}\right)(o_i - \overline{o})}{\sqrt{\frac{1}{N}\sum_{i=1}^{N}(f_i - \overline{f})^2\,\frac{1}{N}\sum_{i=1}^{N}(o_i - \overline{o})^2}} \tag{4}$$

where $N$ denotes the total number of grids in the spatial field, and $f_i$ and $o_i$ indicate the forecasted and observed values for the $i$th grid, respectively. $\overline{f}$ and $\overline{o}$ represent the average values of the forecasted and observed values of all $N$ grid points. The PCC is used to assess the similarity between the observed and forecasted precipitation distributions, with values ranging from $-1$ to 1. A higher PCC value indicates greater similarity, reflecting higher forecast skills.

(4) The index of agreement is obtained from the following:

$$\text{IOA} = 1 - \frac{\sum_{i=1}^{N}(f_i - o_i)^2}{\sum_{i=1}^{N}\left(|f_i - \overline{o}| + |o_i - \overline{o}|\right)^2} \tag{5}$$

where $f$ and $o$ represent the model and observation, respectively. $\overline{o}$ represents the observed mean value and $N$ is the number of total data/grid points. The IOA is bounded between 0 and 1, where a value close to one indicates more efficient forecasting skills [20].

(5) Binary event evaluation metrics are obtained as follows:

Various metrics can be used to evaluate the model forecast results for binary event forecasts with precipitation exceeding a certain threshold. These metrics include the ETS, POD, and FAR, which are employed to examine and evaluate the forecast results of the model.

$$\text{ETS} = \frac{N_A - R(a)}{N_A + N_B + N_C - R(a)} \tag{6}$$

$$R(a) = \frac{(N_A + N_B)(N_A + N_C)}{N_A + N_B + N_C + N_D} \tag{7}$$

$$\text{POD} = \frac{N_A}{N_A + N_C} \tag{8}$$

$$\text{FAR} = \frac{N_B}{N_A + N_B} \tag{9}$$

As shown in Table 4, $N_A$ represents the number of correct forecasts (where both forecasted and observed events occurred), $N_B$ is the number of false alarms (occurrence in forecasts but not in observations), $N_C$ is the number of missing alarms (occurrence in observations but not in forecasts), and $N_D$ is the number of correct rejections (occurrence neither in forecasts nor observations). The ETS is used to assess the forecast skill relative to random forecasting, penalizing both false alarms and missing alarms. Its value ranges from $-1/3$ to 1. Higher ETS values indicate better forecast skills and an ETS value less than or equal to 0 implies no forecasting skill. The POD and FAR have values of [0, 1]. POD refers to the proportion of the predicted actual precipitation area in the total actual precipitation area, and the larger the value, the higher the forecast accuracy. FAR refers to the proportion of the area with no actual precipitation to the total forecast precipitation area; here, the smaller the value, the smaller the forecast null rate. Higher POD and smaller FAR values indicate higher forecast skills.

**Table 4.** Confusion matrix for binary events (forecasts vs. observations).

| | | Forecasts | |
| --- | --- | --- | --- |
| | | **Positive** | **Negative** |
| Observations | Positive | $N_A$ (correct forecasts) | $N_B$ (false alarms) |
| | Negative | $N_C$ (missing alarms) | $N_D$ (correct rejections) |

### 4.2. Spatial Verification

Due to the double-penalty effect of the traditional "point-to-point" statistical evaluation metrics such as the ETS, we also employed the MODE, a spatial verification method, to assess the spatial structural characteristics of precipitation. We aim to provide a more com-

prehensive evaluation and analysis of the forecast results from the two ensemble forecast methods from multiple perspectives.

The MODE method is object-based or feature-based. Initially, it identifies precipitation objects of interest through spatial smoothing and threshold control. To obtain more continuous rainfall regions, the original precipitation field needs to undergo convolution processing in space with a convolution radius $R$ (unit: grid spacing or kilometers), as described by Equations (10) and (11).

$$C(x,y) = \sum_{u,v} \varphi(u,v) f(x-u, y-v) \tag{10}$$

$$\varphi(u,v) = \begin{cases} \frac{1}{\pi R^2}, & u^2 + v^2 \leq R^2 \\ 0, & u^2 + v^2 > R^2 \end{cases} \tag{11}$$

where $f$ represents the original data field, $C$ is the convolution field, and $\varphi$ is the filtering function. $(x, y)$ and $(u, v)$ are the coordinates of the grid points. The focus is solely on rainfall areas with intensity greater than or equal to the threshold value $T$ in the convolution field. Subsequently, threshold control is applied to the convolution field to obtain a mask field $M$.

$$M(x,y) = \begin{cases} 1, & C(x,y) \geq T \\ 0, & C(x,y) < T \end{cases} \tag{12}$$

Finally, the grid points within the continuous region where $M = 1$ are assigned the values of grid points corresponding to the original precipitation field to obtain the reconstructed field $F$. The reconstructed field retains the most original precipitation information for each object (unprocessed rainfall amount), and also, the objects worthy of attention that meet the precipitation threshold are identified.

$$F(x,y) = M(x,y) \cdot f(x,y) \tag{13}$$

Then, the attributes of precipitation objects, predefined and user-selectable, are calculated. For two precipitation objects, one from the forecast field and the other from the observation field, attributes such as the area ratio (ratio of forecasted to observed object areas), axis angle difference (difference in the main axis direction between forecasted and observed objects), overlapping area (area of overlap between forecasted and observed objects) and centroid distance (distance between the centroids of forecasted and observed objects) are evaluated, as shown in Figure 5.

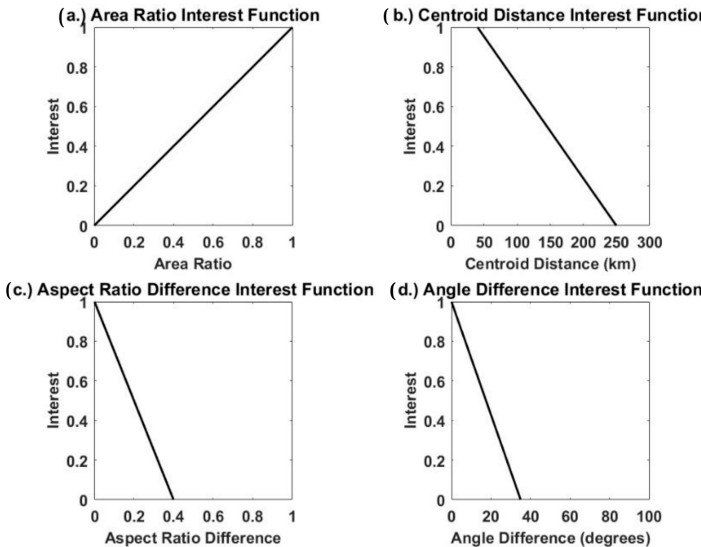

**Figure 5.** (**a**) Area ratio, (**b**) centroid distance and the similarity functions of (**c**) the aspect ratio difference and (**d**) axis angle difference attributes [21].

## 5. Results

### 5.1. Adjustment of Precipitation Ensemble Forecast Fields via the Feature-Oriented Mean Method

Since TC centers among ensemble members gradually deviate from the forecast leading time, TC precipitation in ensemble forecasts also suffers from noticeable location deviations. Moreover, the differences in TC-related dynamical and thermal structures among ensemble members also gradually increase, further contributing to the inter-ensemble variability in TC precipitation fields. Figure 6 illustrates the forecasts of 20 NCEP-GEFS members for the 6 h accumulated precipitation after the landfall of Typhoon Lekima (1909). The landfall time of Typhoon Lekima was at 01:00 (BJT) on 10 August, as indicated in Table 1. The forecast time selected was at 00:00 (UTC) on August 10, and the accumulated precipitation for the preceding 6 h, from 18:00 (UTC) on August 9 to 00:00 (UTC) on 10 August 2019, was used as the prediction. The initial forecast time was at 00:00 (UTC) on 7 August 2019, and a forecast leading time of 72 h was determined. From Figure 6, it can be found that the rainfall area of Member 1 is in southeastern Fujian, while that of Member 20 is in northeastern Taiwan, markedly more eastward and southward compared with Member 1. Members 8, 13, and 14 exhibit weaker precipitation intensity without a pronounced precipitation circulation structure. The forecasted TC regions and precipitation intensities are quite different among different ensemble members. The simple arithmetic averaging ("point-to-point" averaging) of these 20 ensemble members can lead to an overall weakening of precipitation intensity and the excessive smoothing and distortion of precipitation fields.

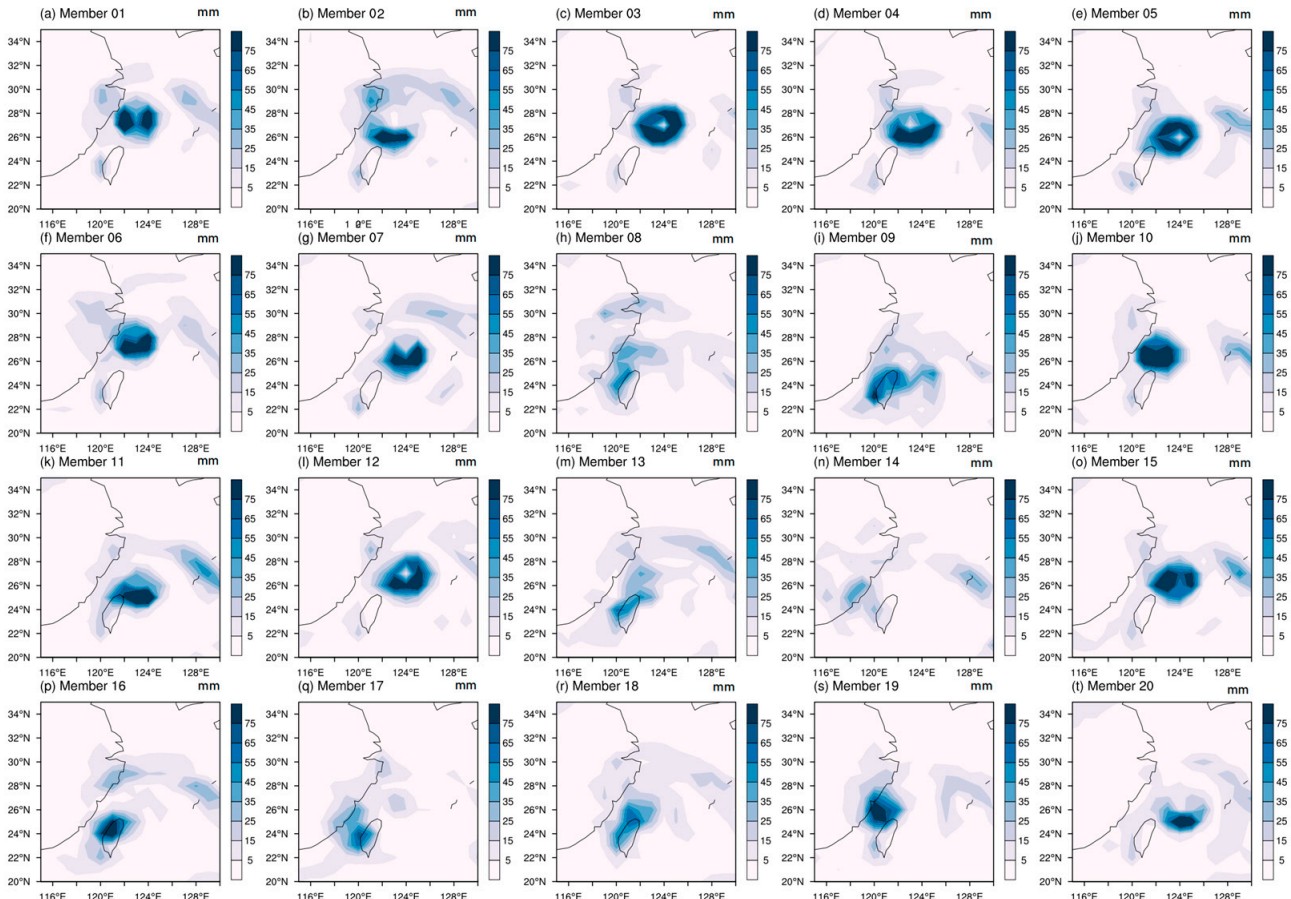

**Figure 6.** The 6 h accumulated rainfall forecasts of 20 GEFS members for Super Typhoon Lekima (1909), with an initial forecast time at 00:00 (UTC) on 7 August 2019 and a forecast leading time of 72 h.

To address the positional differences in the TC precipitation fields among ensemble forecasts in Figure 6, we employed the field adjustment algorithm in the FM method to adjust the inter-ensemble disparities. That is, the precipitation fields of individual ensemble

forecast members were precisely adjusted using the field adjustment algorithm to align with ensemble mean locations, which can mitigate most positional discrepancies in precipitation. Typhoon Lekima is taken as an example in Figure 7 to illustrate how the field adjustment algorithm adjusts the positional features of one precipitation field to another. As shown in Figure 7a, in terms of the 6 h accumulated precipitation forecast fields from 18:00 (UTC) on 9 August 2019 to 00:00 (UTC) on 10 August 2019 for two randomly selected ensemble members, Members 10 and 20 exhibit similar precipitation patterns but have noticeable positional discrepancies in precipitation distribution. The field adjustment algorithm first calculates the displacement field between the two fields. Then, it moves each grid point of Member 10 to Member 20 according to the displacement vector (including displacement direction and distance). From Figure 7b, it can be found that the precipitation distribution of the adjusted Member 10 is closer to that of Member 20. It indicates that the field adjustment algorithm largely mitigates the positional discrepancies of the precipitation fields between these two ensemble forecasts.

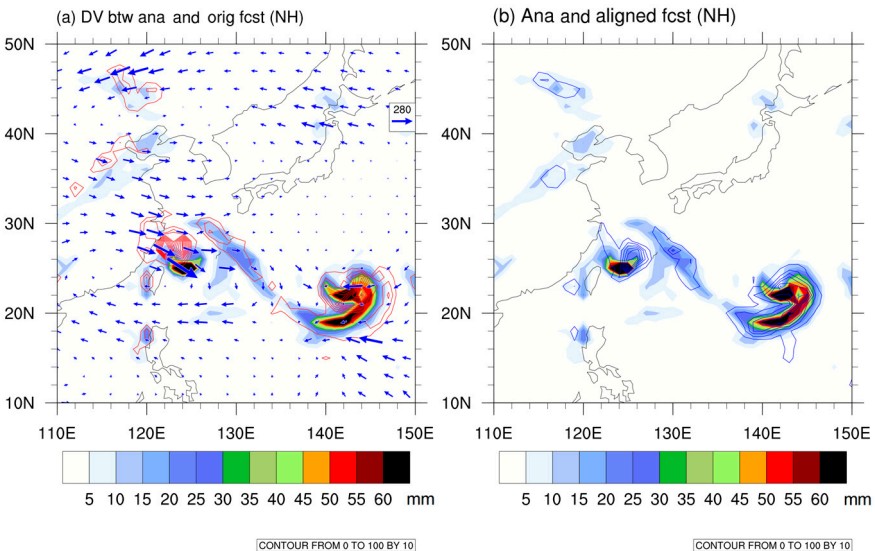

**Figure 7.** The 6 h accumulated precipitation forecasts of two randomly selected ensemble members for Super Typhoon Lekima (1909), initiated at 00:00 (UTC) on 7 August 2019 and a forecast leading time of 72 h. (**a**) Ensemble member 20 (shaded plot), ensemble member 10 (red contour lines), and the displacement vector between them (blue arrows); (**b**) ensemble member 20 (shaded plot) and the adjusted ensemble member 10 after displacement (blue contour lines).

Figure 7 illustrates the process of moving one precipitation field to another based on the field adjustment algorithm. For the GEFS, which included 20 ensemble members, it was necessary to move the precipitation characteristics of each ensemble member to their respective average locations. The determination of the average location and the specific steps are detailed in Section 2.1, or Feng et al. (2020). In this section, the 6 h accumulated precipitation forecast for the Lekima at 72 h forecast leading time from 18:00 (UTC) on 9 August 2019 to 00:00 (UTC) on 10 August 2019 is taken as an example to display the performance of the FM method on the adjustment of the 20 ensemble members. From Figure 8a, it can be seen that the original 10 mm precipitation fields from the 20 ensemble members exhibit large location deviations and a high ensemble spread. In Figure 8b, the FM method is applied when moving each of the 20 members to their mean positions, and the results indicate that this method eliminates most location deviations and makes a more concentrated area of the 10 mm precipitation. Figure 8c,d show the ensemble averages obtained by averaging the precipitation ensemble forecast fields in Figure 8a,b, respectively. It is evident that the FM ensemble forecast field in Figure 8d displays a more pronounced TC precipitation circulation structure compared with the AM ensemble forecast field in Figure 8c. Additionally, the precipitation intensity from the FM, which is closer

to the observation in Figure 8e, is stronger than that from the AM. Similar displacement performance of the FM method can be found in the 48 h and 24 h forecasts of the 6 h accumulated precipitation (from 18:00 (UTC) on 9 August 2019 to 00:00 (UTC) on 10 August 2019) for Super Typhoon Lekima (1909) (figure omitted). These forecasts all show a more distinct TC precipitation circulation structure, and the precipitation intensity of the FM forecasts is stronger than that of the AM forecasts.

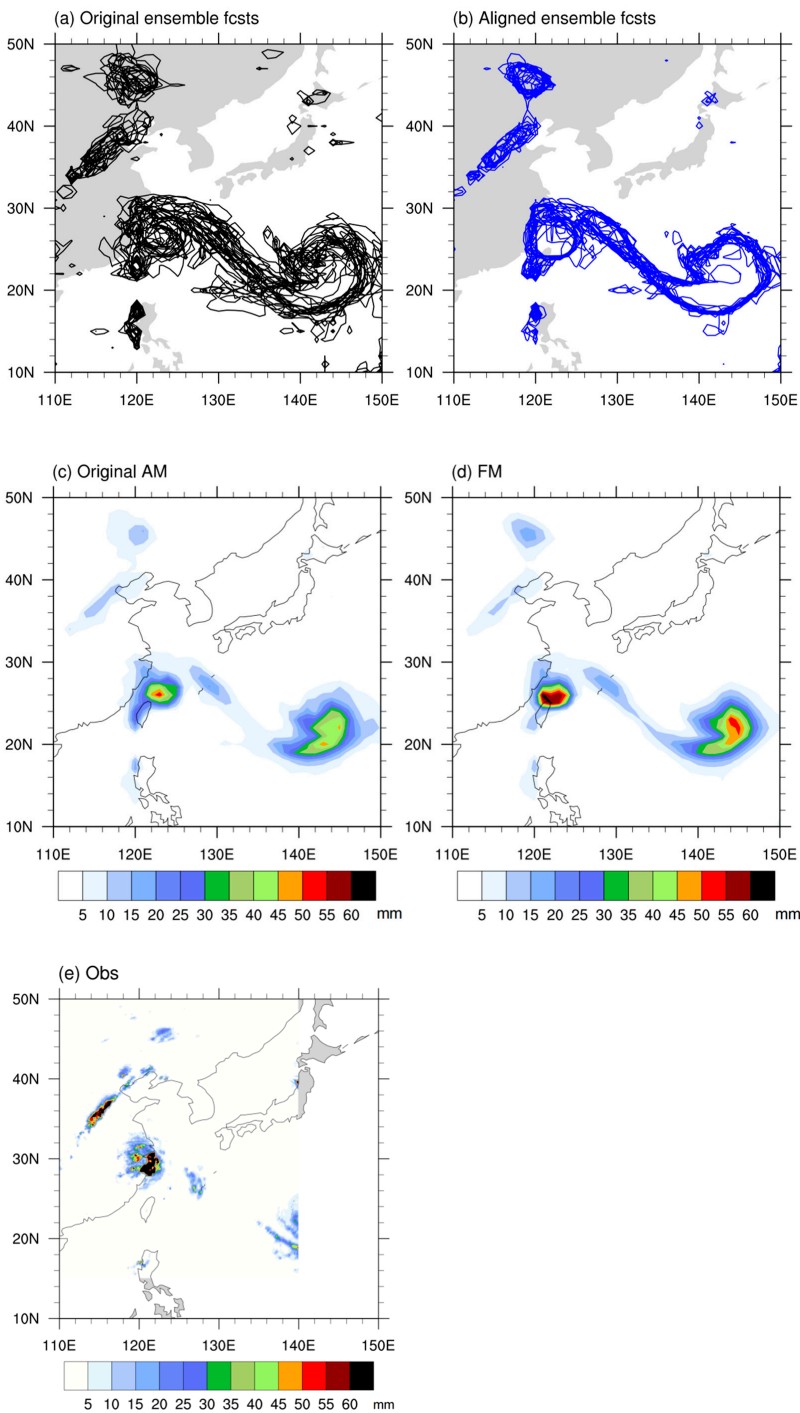

**Figure 8.** The 6 h accumulated precipitation of the GEFS 72 h forecasts for Super Typhoon Lekima (1909): (**a**) original forecasts from 20 ensemble members for 10 mm precipitation, (**b**) 10 mm precipitation fields after FM displacement, (**c**) ensemble average of the original forecast fields for the arithmetic mean (AM), (**d**) ensemble average of the fields after FM displacement, and (**e**) multisource fusion precipitation.

*5.2. Evaluation of the Forecast Performance of the Feature-Oriented Mean Method*

The results in Section 5.1 indicate that the positions of TC precipitation forecast fields for individual ensemble members are more consistent after the FM displacement, and the FM algorithm makes the average ensemble forecast precipitation intensity increase. In this section, we comprehensively evaluate the average ensemble precipitation forecast from the FM method using various indicators, including traditional "point-to-point" rainfall tests and precipitation spatial structure characteristics tests.

To unify the resolution, data from the GEFS model used for the forecast with a coarser resolution (0.5° × 0.5°) were first interpolated into the resolution consistent with this observation (0.05° × 0.05°) using bilinear interpolation. As this study focuses on TC precipitation forecasts, the areas of the same size in the relative coordinates of the TC center are selected for testing in both the evaluation and comparison of the AM and FM methods. Specifically, the TC center positions for the observations, AM, and FM methods were first determined, and a square region of 3000 km × 3000 km was selected around each center as the testing area for TC precipitation. This approach can mitigate the impact of location deviations on precipitation forecast results. Note that the TC center of observations is from the CMA Tropical Cyclone Best Track Dataset. The TC centers for the AM and FM methods were determined by the position of the lowest sea-level pressure in the ensemble-averaged TC vortex fields. As shown in Figure 9, it can be found that the structures of the ensemble-averaged precipitation fields for the AM and FM methods are similar for the 72 h forecasts, mainly because these two methods use the same ensemble forecast members. The same findings can be observed in the 48 h and 24 h forecasts (figure omitted). However, the FM ensemble-averaged precipitation, closer to the observation, is stronger and more concentrated than the AM ensemble-averaged precipitation due to the positional adjustments of the ensemble member features by the FM method. Additionally, the differences between the AM and FM forecasts become more pronounced with the increase in the forecast leading time. This was mainly due to the fact that non-linear effects become more prominent with the forecast leading time, resulting in clearer differences among TC ensemble members and a more noticeable impact of the feature adjustment made by the FM method.

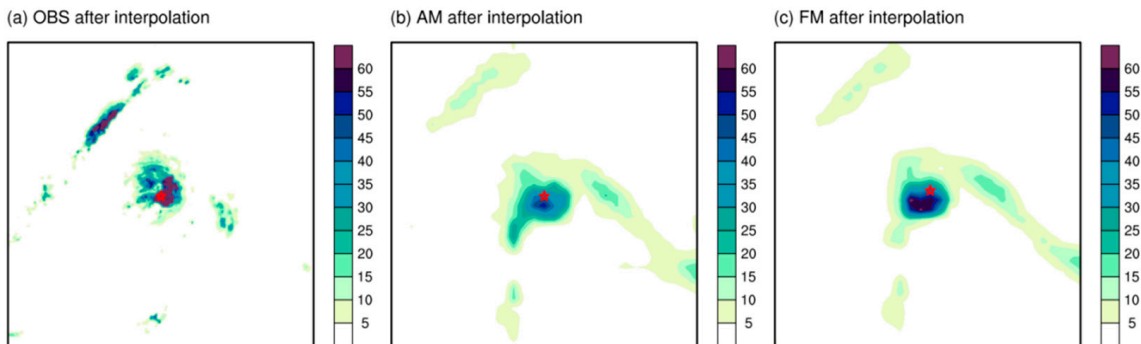

**Figure 9.** Interpolated 72 h forecasts of the GEFS model for the accumulated 6 h precipitation of Super Typhoon Lekima (1909): (**a**) observed precipitation, (**b**) AM precipitation forecasts, and (**c**) FM precipitation forecasts. Red pentagrams represent the TC center.

Furthermore, a binary classification method was used to quantitatively evaluate the precipitation forecast skills of 18 landing TCs in China from 2019 to 2021. The information on each typhoon case is found in Tables 1–3, and the evaluation method is described in Section 4.1. Higher POD, lower FAR, and higher ETS values indicate better precipitation forecast performance. From Figure 10, it is clear that the ETS value of the FM method is higher than that of the AM method for light rain (0.1 mm), moderate rain (4 mm), heavy rain (13 mm), and torrential rain (25 mm) in 72 h, 48 h, and 24 h forecasts. The ETS of the FM method is improved by approximately 10% compared to the AM method. Regarding

the POD, the FM method outperforms the AM method for heavy rain forecasts. In most cases, the FM method also performs better in light, moderate, and torrential rain forecasts. Additionally, the FAR of the FM method is always lower than that of the AM method for light, moderate, and torrential rain forecasts.

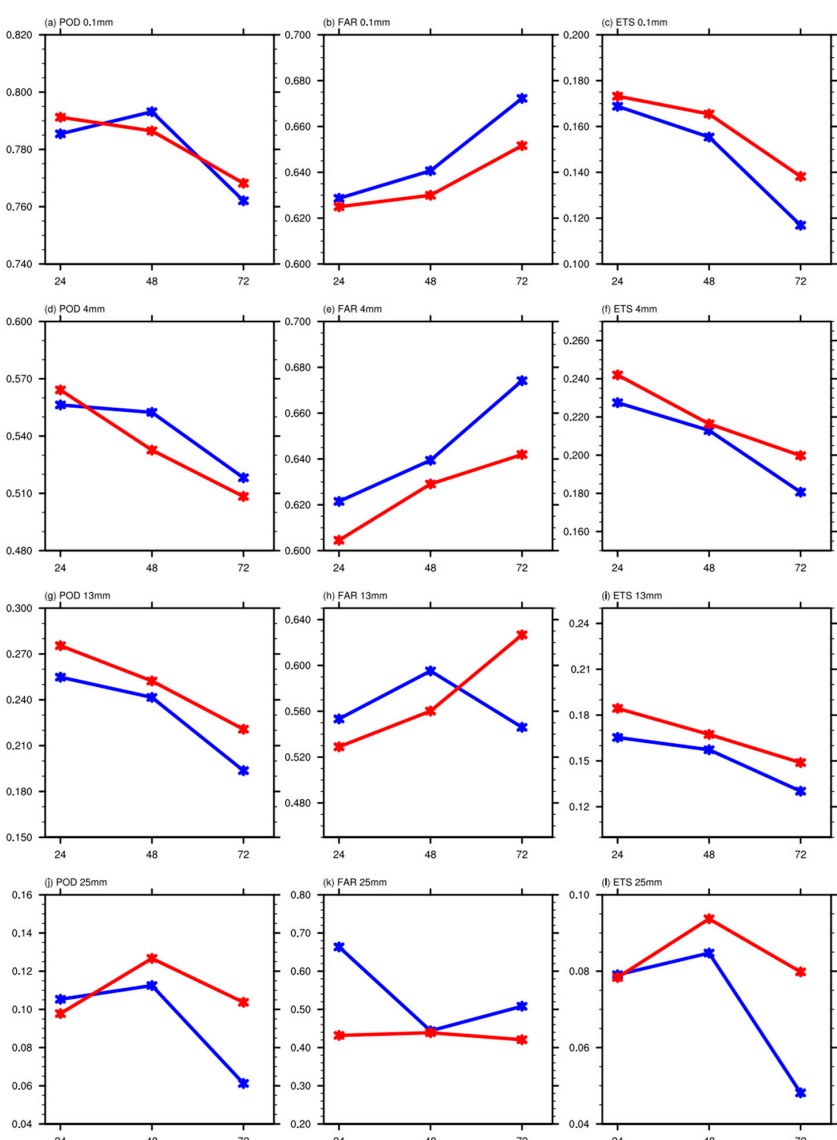

**Figure 10.** The probability of detection (POD), false alarm ratio (FAR), and equitable threat score (ETS) of the 72 h, 48 h, and 24 h forecasts of the AM (blue) and FM (red) methods for the 6 h accumulated precipitation of the 18 landing TCs in China from 2019 to 2021 at grades of (**a**–**c**) light rain (0.1 mm), (**d**–**f**) moderate rainfall (4 mm), (**g**–**i**) heavy rain (13 mm), and (**j**–**l**) torrential rain (25 mm). (24 h, 48 h, and 72 h).

We further utilized the PCC, RMSE, and IOA to evaluate the precipitation forecasts averaged over all cases (Figure 11). The results indicate that the PCC and IOA values of the FM forecasts are larger than those of the AM forecasts, while the RMSE values of the FM forecasts are smaller for different forecast leading times, which suggests that the FM method performs better than the AM method in capturing the overall structural characteristics of the precipitation associated with these landing TCs.

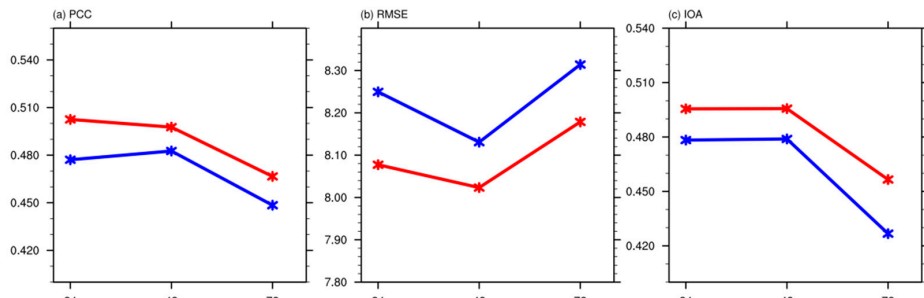

**Figure 11.** The variations in the (**a**) pattern correlation coefficient, (**b**) root mean square error, and (**c**) index of agreement for the 72 h, 48 h and 24 h forecasts of the AM (blue) and FM (red) methods for the 6 h accumulated precipitation of the 18 landing TCs in China from 2019 to 2021.

The above traditional "point-to-point" rainfall tests and the tests focused on precipitation structure indicate that the ensemble average precipitation forecast skill of the FM method is higher than that of the AM method. Additionally, in this section, we also employed the MODE to examine the spatial characteristics of the precipitation structure. The goal was to comprehensively evaluate and analyze the capabilities of both ensemble forecasting methods to capture the spatial structural features of TC precipitation. The details on the MODE test method are in Section 4.2.

In the MODE spatial test, a convolution radius of four grid distances was employed for spatial smoothing. Precipitation thresholds of 4 mm and 25 mm were selected. The attributes of the precipitation objects identified by the observations and two ensemble forecast methods are shown in Figures 12 and 13. It can be seen that for both 4 mm and 25 mm precipitation, the attribute distribution of the FM forecast objects is more similar to that of the observed precipitation objects in terms of the area, longitudinal centroid position, latitudinal centroid position, axis angle and aspect ratio. In other words, the FM method performs better than the AM method in capturing the characteristic attributes of precipitation.

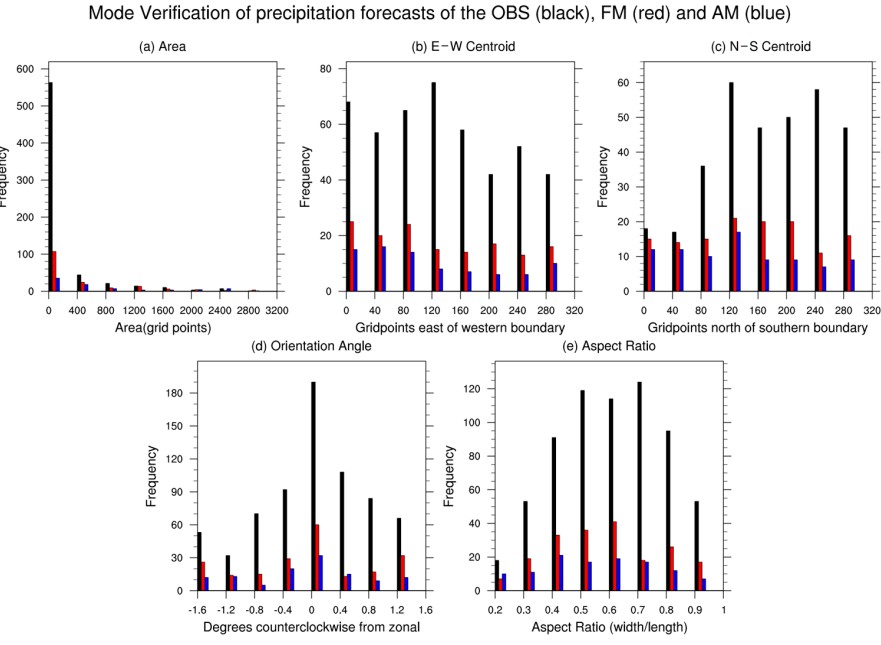

**Figure 12.** Attribute distribution of the observations (black, OBS) and 72 h forecasts from the FM (red) and AM (blue) for the 6 h accumulated precipitation of 18 landing TCs from 2019 to 2021 at the 4 mm threshold: (**a**) area (unit: grids), (**b**) longitudinal centroid location, (**c**) latitudinal centroid location, (**d**) axis angle, and (**e**) aspect ratio.

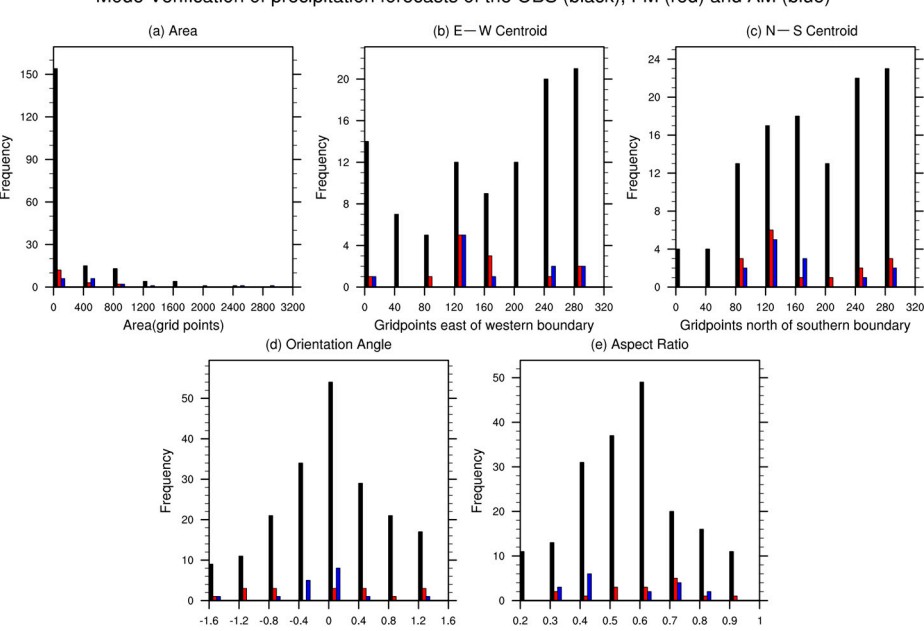

**Figure 13.** Same as Figure 12 but for the 25 mm precipitation threshold.

## 6. Conclusions and Discussion

Location deviations are noticeable in TC precipitation among ensemble forecast members due to the deviations in the TC center locations among ensemble members and the differences in TC structures. The widely used AM method does not consider the location deviation of precipitation among ensemble members, resulting in the over-smoothing of the TC ensemble mean precipitation structure and a remarkably weaker intensity. Effectively estimating the ensemble average of TC precipitation is a critical and forefront issue in TC forecasts globally. In this study, an ensemble mean algorithm, FM, is developed based on structural features suitable for spatially discrete variables in precipitation ensembles. This method involves adjusting the positions of the precipitation fields to reduce the location deviations among ensemble members, followed by averaging precipitation intensities. This enhances the ensemble forecast skill of TC precipitation. To validate the feasibility of the FM in TC precipitation ensemble forecasts, we selected 18 landfalling TC cases in China from 2019 to 2021 in this research. The precipitation forecast skills of the FM and AM algorithms were quantitatively evaluated and compared comprehensively at different leading times. The main conclusions are discussed as follows.

The FM field adjustment algorithm can effectively adjust the TC precipitation structure to the ensemble mean position, reducing the spatial dispersion among precipitation fields. Consequently, the ensemble-averaged precipitation field after the FM displacement (closer to the observations) exhibits a more concentrated precipitation structure and higher precipitation intensity compared with the AM forecasts.

Through the "point-to-point" verification of the 6 h accumulated precipitation forecasts of 18 landing TCs between 2019 and 2021, it was found that for the ETS, POD, and FAR, the FM method outperformed the AM, with an overall improvement of around 10%. Additionally, compared with the AM ensemble mean, the FM ensemble mean showed a higher PCC value with observed precipitation and a smaller RMSE value, which meant that the FM could better preserve the structural characteristics of precipitation. The MODE verification of the forecast results also indicates that the attribute distribution of the FM forecast objects is clearly more similar to that of observed precipitation objects, whether in terms of the area, longitudinal centroid location, latitudinal centroid location, axis angle, or aspect ratio.

Nonetheless, future breakthroughs and advancements in TC precipitation ensemble forecast skills are still needed in several aspects.

Firstly, there is a need for the refinement of the FM algorithm. Currently, the FM algorithm used involves adjusting the locations of all ensemble forecast members and then directly conducting an equally weighted ensemble mean forecast. It is noteworthy that some ensemble members may not reflect the actual state of the atmosphere in operation and may show large deviations. These outliers or low-probability events carry certain statistical significance, but they can reduce the overall accuracy of ensemble forecasts. Therefore, when the FM method is used for ensemble forecasting, exploring the possibility of sample selection among the ensemble members, or assigning different weights to different ensemble members before ensemble averaging could potentially enhance FM forecasting skills.

Secondly, the development of high-performance computing should be taken into consideration. Due to limited computational resources, the number of TC ensemble forecast members is typically between 20 and 30 in various operational units. Even with the top-notch computing conditions in the European Centre for Medium-Range Weather Forecasts, only around 50 ensemble forecast members are used. Moreover, the resolution of ensemble forecast systems is noticeably lower than that of control forecasts. It prevents ensemble member forecasts from effectively distinguishing the convective-scale structure of TCs, forming a major obstacle to the high-resolution development of TC forecasts. Improving the computational efficiency and performance of computers to increase model resolution and the number of ensemble members in TC ensemble forecasts requires collaborative efforts from experts in computer science, meteorology, and other related fields.

Moreover, there is a need for the development of post-processing methods for TC ensemble forecasting. Currently, there are various methods for post-processing ensemble forecasts, such as the deviation correction-based method, Bayesian theory-based method, and clustering analysis-based method. Combining cutting-edge machine learning, post-processed TC ensemble forecast data, and conducting intelligent forecasting represents an important direction for the development of TC ensemble forecasting.

**Author Contributions:** Conceptualization, J.Z.; methodology, J.Z.; software, J.Z.; validation, J.Z. and H.L.; formal analysis, J.Z. and H.L.; investigation, J.Z. and H.L.; resources, J.Z.; data curation, J.Z. and H.L.; writing—original draft preparation, J.Z.; writing—review and editing, H.L.; visualization, J.Z. and H.L.; supervision, H.L.; project administration, J.Z. and H.L.; funding acquisition, J.Z. and H.L. All authors have read and agreed to the published version of the manuscript.

**Funding:** This study was supported by the National Natural Science Foundation of China (42305060) and the Program of Shanghai Academic/Technology Research Leader (21XD1404500).

**Data Availability Statement:** The raw data supporting the conclusions of this article will be made available by the authors on request.

**Conflicts of Interest:** The authors declare no conflict of interest.

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
