# Peer review of "Application of Ensemble Algorithm Based on the Feature-Oriented Mean in Tropical Cyclone-Related Precipitation Forecasting"

_remotesensing, doi:10.3390/rs16091596_

Round 1

Reviewer 1 Report

Comments and Suggestions for Authors

Summary:

This study provides an alternative approach to the calculation of ensemble mean of precipitation forecasts, which is named as the feature-oriented mean. This FM approach reduces the positional deviations across ensemble members, resulting in an improved ensemble mean performance. The content, objectives and key scientific issues of this manuscript are reasonable and innovative. The main results are promising. The paper is generally well written. Consequently, it is recommended to accept this article after a minor revision.

General comments:

(1) The manuscript examines the performance of FM in precipitation ensemble forecasts using the Global Ensemble Forecast System (GEFS) product at NCEP . My concern is that the GEFS data has a relatively low resolution. It is necessary to give more discussion on whether the FM would still be valid and useful for the precipitation forecasts in regional high-resolution models.

(2) The description of the verification metrics including ETS, POD, FAR were not clear enough. Please improve the introduction of these metrics.

(3)  Some previous research uses the relocation scheme to shift the TC structure in individual members to the ensemble mean position of their vortices. What is the difference between the relocation scheme and the FM approach? Would they perform similarly in the ensemble mean forecast of precipitation?

(4)  The spaghetti plot (Fig. 10) is not clear. Please replot the figure.

(5)  Latest references about tropical cyclone ensemble forecast should be included in the introduction parts to show this study’s importance and new advances.

References:

1. Lu D, Ding R, Mao J, et al. Comparison of different global ensemble prediction systems for tropical cyclone intensity forecasting[J]. Atmospheric Science Letters, 2024: e1207.

2.Zhou X, Zhu Y, Hou D, et al. The development of the NCEP global ensemble forecast system version 12[J]. Weather and Forecasting, 2022, 37(6): 1069-1084.

Comments on the Quality of English Language

The paper is generally well written.

Author Response

Thanks for your good suggestions. Please see the attachment.

Reviewer 2 Report

Comments and Suggestions for Authors

General comments: 

 This study applied the feature-orientated mean (FM) approach to ensemble forecasts of typhoon precipitation. FM adjusts the location of precipitation in individual ensemble fields to reduce their positional deviations across members before averaging the members. The FM has the potential to be an alternative to the widely used algorithm mean. The manuscript was generally well-written and logically organized. I would recommend a minor revision before the manuscript can be considered for publication. 

Minor comments:

(1)  I notice that while the alignment of ensemble members adjusts the positions of precipitation, it also causes some stretching and deformation of the structure of precipitation. Is it true and what influence it may have on the FM?

(2)  I believe the FM may not guarantee the conservation of air mass, kinematics, and thermodynamics. Would that be a problem?

(3)Is the calculation of FM time-consuming? If the algorithm is computationally expensive, it will constrain its application to operations. In particular, the operational TC forecast systems.

Comments on the Quality of English Language

The paper is well-written with smooth sentences, and the choice of words is appropriate. 

Author Response

(The authors gave the same response as above.)

Reviewer 3 Report

Comments and Suggestions for Authors

Thank you for considering me as a reviewer of the manuscript. I found the manuscript to be interesting and relevant to the existing literature on Tropical Cyclone-Related Precipitation Forecasting. The authors have used the feature-oriented mean (FM) method to adjust the locations of ensemble precipitation fields to reduce the location-related deviations among ensemble members. Some suggestions are:

1. Is it possible to use this method to forecast future TCs? Or this method is used only to adjust the locations of ensemble precipitation fields.

2. Is this method suitable only for adjusting precipitation fields or could we use this method to adjust all other atmospheric fields?

3. I would suggest you to use index of agreement (IOA) instead of correlation coefficient to measure the accuracy of precipitation fields. Correlation coefficient is often deceptive and are not related directly to the accuracy of prediction. You can find the formula and details about this method in this paper.

Shahi, N. K., Polcher‬, J., Bastin, S., Pennel, R., & Fita, L. (2022). Assessment of the spatio-temporal variability of the added value on precipitation of convection-permitting simulation over the Iberian Peninsula using the RegIPSL regional earth system model. Climate Dynamics, 59(1), 471-498. https://doi.org/10.1007/s00382-022-06138-y.

Comments on the Quality of English Language

Minor editing of English language required

Author Response

(The authors gave the same response as above.)

Reviewer 4 Report

Comments and Suggestions for Authors

The manuscript deals with an ensemble forecast of tropical cyclones to improve the accuracy of rainfall forecasts using a three-year landfall typhoon in China through FM(feature-oriented mean). The authors organized the manuscript well to explain the background and results. However, it remained to solve something to be accepted. I would like to recommend that it would be accepted after major revision.

1. Figure 7: The unit of legend(shading) would be inserted like 'mm'.

2.  Lines 404-406: For comparison of rainfall between AM and obs, FM and obs, it looks that AM has a better match to observation than that of FM. Are there any comments on this?

3. Figure 8: The unit of legend(shading) would be inserted like 'mm'. The authors need to make the color table range same for better understanding in Figures 7 and 8. And the authors make the domain of the figure of (d) and (e) same.

4. Lines 449-460: I would like to know how many point data(obs. data) used for the verification. and how to do verification. I wondered that the authors used both position and intensity of rainfall for the verification.

5. Figure 11: We usually expect that the longer the forecast, the worse the accuracy. But, in this figure, 48 hours has better RMSE performance than 24 hours. Any idea about this?

Author Response

(The authors gave the same response as above.)

Round 2

Reviewer 3 Report

Comments and Suggestions for Authors

I agree with the author's responses, however, I suggest the author to cite this paper for the Index of agreement index.

Shahi, N. K., Polcher‬, J., Bastin, S., Pennel, R., & Fita, L. (2022). Assessment of the spatio-temporal variability of the added value on precipitation of convection-permitting simulation over the Iberian Peninsula using the RegIPSL regional earth system model. Climate Dynamics, 59(1), 471-498.

Comments on the Quality of English Language

Minor editing of English language required

Author Response

Thanks for your good suggestions. We have cited this paper for the IOA in this paper (line 296 and lines 615~617).

Reviewer 4 Report

Comments and Suggestions for Authors

I think that the authors modified the manuscript according to the comments. However, there are unknown characters in Equations. They should be corrected before publication.

Therefore, I would like to recommend that it would be accepted after minor revision.  

Author Response

Thanks for your good suggestions. For clarity, we have rewrote the following explanation for equation 1(line 150~162), equation 3 (line 279~281) and equation 5 (line 293~294).